# Comparative Levelized Cost Analysis of Transmitting Renewable Solar Energy

Clinton Thai 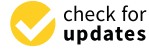 and Jack Brouwer *

Advanced Power and Energy Program, University of California Irvine, Irvine, CA 92697, USA
* Correspondence: jb@nfcrc.uci.edu

**Abstract:** A bottom-up cost analysis for delivering utility-scale PV-generated electricity as hydrogen through pipelines and as electricity through power is undertaken. Techno-economic, generation, and demand data for California are used to calculate the levelized cost of transmitting (LCOT) energy and the levelized cost of electricity (LCOE) prior to distribution. High-voltage levels of 230 kV and 500 kV and 24-inch and 36-inch pipelines for 100 to 700 miles of transmission are considered. At 100 miles of transmission, the cost of transmission between each medium is comparable. At longer distances, the pipeline scenarios become increasingly cheaper at low utilization levels. The all-electric pathways utilizing battery energy storage systems can meet 95% of the load for as low as 356 USD/MWh, whereas when meeting 100% of load with the hydrogen gas turbine and fuel cell pathways, the costs are 278 and 322 USD/MWh, respectively.

**Keywords:** hydrogen transmission; hydrogen storage; large-scale solar integration; electricity transmission; cost analysis

## 1. Introduction and Background

A review of building electric transmission lines by Eto [1] addresses the challenges of siting and evaluating the value of large transmission projects involving various agencies and stakeholders. An evaluation of what the "most efficient" method is for delivering electricity from a point A to point B does not capture the potential needs of stakeholders. The idea of linepack for renewable gases, such as substitute natural gas or renewable hydrogen has inherent value that is rarely quantified [2–4] for cost analyses. Doing so fairly would need to account for renewable goals, the generation dynamics of other states, the benefits of other sectors outside of power generation (e.g., transportation), public opinion, safety, and the reliability of transmission infrastructure.

The California Energy Commission has mandated solar installations for new homes built in California to be sized to produce as much electricity as their annual consumption [5]. Doing so has major implications on the energy system. This will increase the need for controllable resources, such as dispatchable energy storage. In addition, fewer centralized power plants will be needed—likely reducing the midday utilization factor of te transmission lines. On a local level, even a slight 5% blend of hydrogen in Southern California's existing natural gas pipelines would provide 650 GWh of energy storage equivalent to USD 130 billion in battery costs [6], potentially alleviating transmission constraints arising from solar-project-populated Central California and enabling further solar deployment.

With potentially large infrastructure costs, the most economic method of balancing generation and load is of great interest. The cost of transmitting energy throughout society is dependent on numerous factors. Two notable factors are transmission distance and the utilization factor of the transmission medium [7]. In addition, the levelized cost of transmission (LCOT) can have significant differences depending on the voltage levels and conductor sizes in the electricity scenario. Pipeline diameter and pressure are factors in the gaseous scenario. Some transmission planning projects focus on relieving congestion,

whereas other projects are system capacity expansions. The geographical location of transmission infrastructure adds another layer of specificity as construction costs and right-of-way costs may vary by terrain type. A work from Mills et al. [8] is a comprehensive review of transmission planning studies and an attempt to relate the investment costs with the new wind generation they support, acknowledging many of the possible geographical integration differences. Kishore and Singal [9] also highlight the economic discussion of electric transmission lines in their review, but their work primarily focuses on providing the framework for discussing and analyzing transmission costs rather than conducting numerical studies.

Because of the range of specificity of scenarios, the electric transmission investment costs rather than levelized costs are more readily available [10]. Published capital cost numbers are typically provided for specific projects or are best estimates based on historical or planned transmission assets. For example, capital cost values are typically from regulatory agencies [11–14] and from independent system operators [15,16]. Additionally, some studies like that from Dismukes et al. aim to develop empirical cost models for electric power lines [17].

The levelized cost of electricity (LCOE) from renewables is already widely explored as major project developments have occurred in the past decade. Findings of energy storage enabling further deployment [18] have motivated many more works that focus on the levelized cost of renewable electricity [19–21] and storage [22,23]. The EIA published a report which estimates the levelized cost of production, storage, and upgrading spur lines [24], but does not seem to capture the necessary bulk transmission system upgrades costs. While much work aims to quantify the potential levelized cost of renewable energy with storage, cost analyses that explore the integrated LCOT are hugely lacking.

For hydrogen, the cost of transmission is dependent not only on the capital expenses, but also the operating expenses associated with compression. A 1993 study by Oney et al. considered the cost of transmitting energy as hydrogen at different volumetric blend fractions, transmission distances, and diameters [7]. They found that hydrogen has higher operational costs as its lower volume energy density requires more compression work. However, when feeding compressors hydrogen from higher electrolyzer outlet pressures, less compression is necessary resulting in lower costs than natural gas [7]. Yang and Ogden [25] analyze the best transmission and distribution mode for delivering hydrogen to fueling stations and use an adapted assumption of pipeline costs solely dependent on diameter size from Parker [26]. Note that hydrogen pipelines are far less prevalent than electric power lines, and the best empirical model for pipeline costs to-date seems to be Parker's regression of historical natural gas pipelines costs [26]. The results of Parker's work are also used in the United States Department of Energy's (DOE) Hydrogen Delivery Scenario Analysis Model (HDSAM) [27].

Many analyses exist that attempt to quantify the levelized cost of producing hydrogen from renewables for industrial uses, transportation fuel, and power generation [25,28–31]. Of these, Kluschke and Neumann [31] is the only work that accounts for the geographical differences in feedstock electricity price for different demand locations which could be used to compare to alternative hydrogen transmission costs. Weidner et al. account for the transmission cost of feedstock electricity by considering region-specific grid fees along with other hydrogen production costs, deducing that the resulting levelized cost can significantly vary based on these fees [29]. In general, most levelized costs of hydrogen analyses consider utilizing grid electricity at a fixed cost, which (1) fails to capture the dynamics and congestion associated with renewable generators, and (2) assumes bulk energy transmission as electricity without a comparable centralized hydrogen production and transmission scenario.

Regarding renewable power generation, much more works exist in the literature which investigate the LCOE and storage rather than the transmission aspect. Despite transmission being a much smaller cost than generation and distribution, system upgrades haled by increases in renewable generation seem inevitable and integrated analyses are warranted.

This paper compares hydrogen and electricity transmission pathways when implementing storage to complement solar PV. The levelized cost of delivered renewable electricity is calculated. The effect of varying transmission lengths and generation scenarios is analyzed to explore how total costs may change in a renewable future landscape.

## 2. Approach

Section 2.1 provides a general system description and elaborates on the overall study design logic in comparing the all-electric pathway and the gas pathway, before detailing the major components unique to the all-electric pathway in Section 2.2, and th emajor components unique to the hydrogen pathway in Section 2.3.

### 2.1. System Description and Logic

A point-to-point model is developed with multiple zero-dimensional models in series (see Figures 1 and 2) to represent each major energy process conversion in delivering wholesale utility-scale electricity from one region to another. The primary transmission pathways considered in this work are an all-electric pathway as well as a hydrogen pathway which utilizes transmission of energy as gas through pipelines. Multiple comparisons are made throughout this work by evaluating the two pathways under equivalent bases with the overarching theme being: (1) transmitting energy in general, most akin to meeting heating demands, and (2) transmitting energy for the explicit purpose of meeting end-use electric demand (which requires re-electrification in the gas pathway).

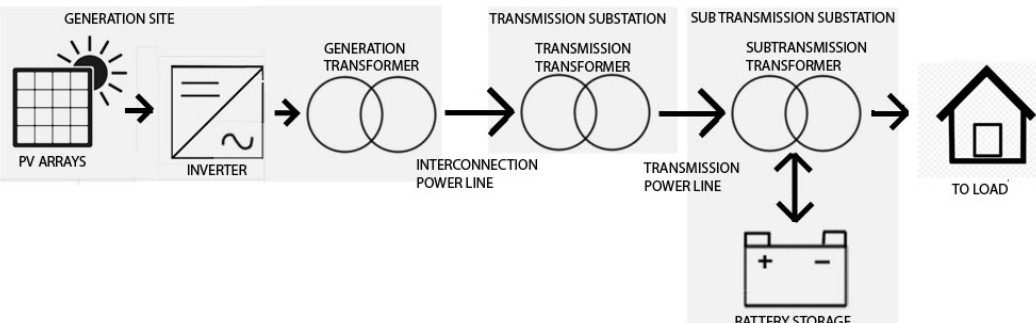

**Figure 1.** Electric pathway considered. Battery energy storage is assumed to have the necessary power conditioning units. The battery system interconnection line is assumed to have negligible costs and efficiency losses.

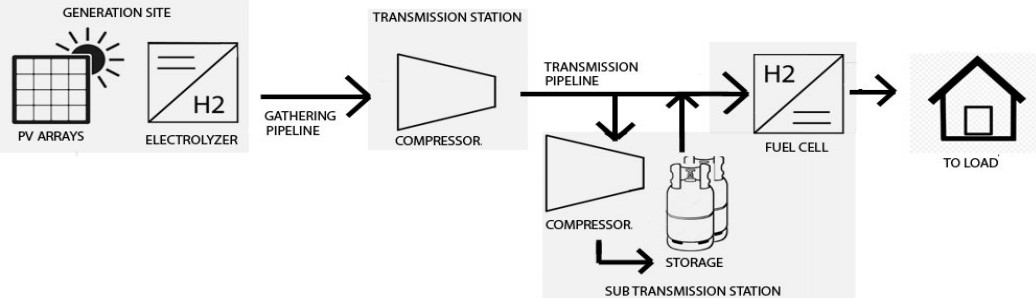

**Figure 2.** Hydrogen pathway considered. Gathering pipeline is assumed to have negligible costs and efficiency losses. In gas turbine scenarios, the fuel cell is replaced. Fuel cells are modeled with necessary power conversion units.

The capacities of components are sized proportional to upstream component sizes and process efficiencies, constrained by the same bases. To equivalently compare the two different transmission media and the cost of transmitting energy generally (i.e., LCOT), the utilization factor of the transmission medium is used as the basis for comparison. When

evaluating the transmission of energy to explicitly meet electric load (i.e., LCOE) multiple bases are considered: (1) the equivalent utilization of the transmission medium, (2) meeting only 95% of the load assuming other resources are available, and (3) meeting 100% of the load to guarantee renewable primary energy.

The evaluated all-electric pathways include a Li-ion battery system and comparable hydrogen pathways are presented with underground geological storage. Underground storage is modeled after depleted natural gas and oil fields, and pipeline linepack storage is not initially considered as an unusable asset. This assumption is inconsequential since the pipeline linepack is innate and saves the need of implementing underground storage and is explored thereafter. Major contributors to energy losses and costs are considered and characterized by information found in the literature and existing analysis tools.

We define the demand end as a sub-transmission station representing the city gate and the generation point as a solar photovoltaic (PV) farm, as shown in Figures 1 and 2. If Los Angeles city is the delivery point, a 100-mile scenario is representative of a PV site located in Riverside County, a high solar resource area; a 500-mile scenario would be representative of transmitting energy from New Mexico state; and a 900-mile scenario is the distance of electricity imports from the Pacific Northwest region, as well as the distance natural gas is imported from Texas. Modeling the longer distances reflects interstate exchanges, such as western states sending excess solar-produced electricity to the east during peak generation, and Midwestern states sending excess wind-produced electricity the other way. This becomes more likely as states continue to construct new solar and wind farms to meet renewable goals [32]. For both pathways, a minimum of two substations are considered in the pathways: transmission and sub-transmission—one effectively at the start of the bulk transmission lines or pipeline and one at the other end (each with their own icon as seen in Figures 1 and 2).

### 2.2. All-Electric Pathway Assumptions

The output voltage for PV power plants is in the 6–36 kV range before being converted to higher voltage levels for transmission [33]. This is supported in the literature where an existing PV farm in India uses a 380 V/33 kV AC transformer [34], and a proposed farm in Libya would utilize 400 V/11 kV [35] for this step. If electricity is delivered locally, these voltage levels are sufficient. However, for long-distance delivery, electricity is routed to a nearby transmission substation (see grouping in Figure 1), where it is converted to higher voltage levels. In California, 115 kV and 230 kV are the most common intrastate transmission levels, and 500 kV is commonly used for interstate exchanges [36]. The National Renewable Energy Laboratory models the cost of PV generation systems and indicates the inverter for a 100 MW system accounts for 5.4% of the PV plant LCOE [37], so that the hydrogen scenarios, which use DC, discount this amount with the omission of the inverter (see Figure 2). The California Public Utilities Commission (CPUC) requires electricity utility companies to provide interconnection cost estimates for new generators. These cost guides list 230 kV and 500 kV as the bulk transmission voltage levels. Cost estimates found in Southern California Edison's (SCE) 2018 cost guide values [13] are used. The O&M costs for electrical lines are estimated by considering SCE's forecasted 2018 O&M costs for transmission lines and substations.

#### 2.2.1. Power Lines

Most transmission lines in Southern California outside of major cities are single-circuit 230 kV and upon entering the city, are converted to the sub-transmission level of 66 kV [36]. Double-circuit lines are modeled as an economical solution to the anticipated transmission capacity constraints.

Power line losses are dependent on the total current and resistance calculated by Ohm's law. The electrical resistance and power rating of the transmission line varies based on the conductor. Many SCE 230 kV transmission upgrades [38] and new 230 kV power lines [39] use 1590 aluminum conductor steel-reinforced (ACSR) cables, one of largest

listed in the ACSR datasheets [40]. The resistance per length for 1590 ACSR is assumed to be constant at 25 degrees Celsius, corresponding to 0.0359 $\Omega$/km. New power lines are assumed to be built in the same areas as existing lines, so right-of-way (ROW) costs are assumed to be negligible. The efficiency of the power lines is determined by calculating power dissipation according to Ohm's law.

### 2.2.2. Transformers

As the transmission lines approach consumers, voltage levels are typically converted back to medium voltage levels for sub-transmission (see Figure 1). Based on existing substations, a 230 kV/66 kV transformer is typical [41]. An additional 500 kV/230 kV transformer is considered for the 500 kV scenario.

Transformer efficiency varies based upon whether the transformer is operating near its rated load [42]. In scenarios where they operate near the rated load, higher efficiencies can be expected, whereas at lower loads the efficiency drops. Accounting for transformer operational dynamics and thermal impacts are avoided in this work due to the level of complexity that would be required to account for these operating characteristics. A 2014 analysis regarding power-to-gas storage complementing wind considered a minimum transformer efficiency of 80%, a maximum of 98%, and a base scenario of 95%. Zini and Rosa [43] model and validate an Italian PV system and also use a 98% transformer average efficiency. If serving PV loads, the transformer can be shut off outside of the predictable generation times—reducing idle no-load losses. Considering this, an average transformer efficiency of 97% is assumed in this work.

Transformer costs are challenging to estimate as they are specifically designed for certain applications and subject to a complex procurement process [44]. A 1997 study by Dagle and Brown [45] found that transformer costs can be estimated as a function of capacity and higher-side voltage levels—estimating USD 7.6 million for a 280 MVA, 230/66 kV transformer. Black & Veatch has evaluated capital costs for electricity transmission infrastructure for the Western Electricity Coordinating Council (WECC) [14]. In addition, the SCE cost guides have some transformer cost estimates. Ultimately, the average USD/MVA of the three above studies for a 230/500 kV is used, with a higher weight assigned to the SCE number (1.5:1:1) due to its geographical pertinence. These cost models are adjusted for inflation to 2017 USD.

### 2.2.3. Battery

The electric energy storage systems (ESS) typically have their own power conditioning systems and controllers which are assumed to be lumped into the system (see Figure 1) cost and efficiency. The parasitic losses are included in the roundtrip efficiency [46]. Lazard's levelized cost of storage analysis provides valuable insights regarding the typical costs of different ESS types. Battery system costs in this work are set to 350 USD/kWh and the corresponding inverter costs are set to 80 USD/kW [47]. Customer-led network revolution implemented a Li-ion-based ESS to support a primary transformer [46]. Its roundtrip efficiency is 69.0% when considering all conversion and balance of plant losses. A summary of the major components is illustrated in Figure 1 for the all-electric pathways with each labeled symbol modeled with an average efficiency and cost numerically tabulated in Table 1.

### 2.3. Hydrogen Pathway Assumptions

Hydrogen gas is assumed to be produced at the PV site, delivered at the electrolyzer outlet pressure (435 psia) to a transmission station, where it is compressed to 1500 psia (see grouping in Figure 2). It is assumed that the pipeline between the PV plant and compressor station is short and does not contribute to total pathway cost and efficiency, which is analogous to disregarding the electrical equipment costs to the transmission station. The electrolyzer outlet pressure is analogous to the medium voltage levels preceding long distance transmission. In the all-electric scenario, inverters and generation transformer

losses are considered at the generation site and for the hydrogen scenario, the losses from the boost converter and electrolyzer are considered.

### 2.3.1. Transmission Pipeline

The pipeline is not spatially resolved, rather it is assumed to comprise one long control volume between each compressor station. The maximum pressure inlet, minimum pressure outlet, and other factors considered in the Darcy–Weisbach equation are used to determine the pipeline hydraulic capacity [48]. The Colebrook–White correlation is used to determine the frictional losses that must be overcome by the compressor stations. When an additional pipeline is necessary, the flow rate is split so that the capacity and utilization factor of each of the pipelines are equal. The same is done in the electric scenario.

Because pipeline pressure drop is a function of length and throughput, one can install compressor stations in series to recover pipeline pressure or install parallel pipelines to reduce the flow rate per pipeline. A constraint is set to have additional compressor substations every 150 miles and additional parallel pipelines are only installed if still necessary. If the transmitted renewable energy is more than enough to meet the demand load, then the surplus is sent to storage. The amount of energy sent to storage is directly related to the available amount of energy after transmission.

The downstream end of each pipeline also houses a fuel cell to allow a comparable analysis for serving electric loads (see Figure 2) where hydrogen is fed in at a pressure of 500 psia. An alternative pathway is developed which considers using a hydrogen gas turbine rather than a fuel cell. A 2004 study takes into account historical pipeline costs to develop an empirical cost model [26]. Parker [26] suggests a multiplier of 1.5 for material costs to address hydrogen embrittlement, 1.25 for a lack of skilled labor regarding these pipelines, and 1 for the miscellaneous category. However, these multipliers are arbitrarily determined in both Parker's work [26] and in the DOE's model, which assumes multiplier values of 1.1 for each factor [27]. A 2015 work conducts a cost analysis to evaluate the pipeline thickness necessary to transport, based on the ASME hydrogen pipeline code to better understand the material cost [49]. In this work, Fekete et al. find that a technical based proposed adaption to the ASME code can reduce pipeline costs by as much as 31%, relative to natural gas pipelines for a 24-inch diameter pipeline operating at 1500 psia. In this work, it is assumed that the developers for hydrogen pipelines will be the same owners as natural gas pipelines, so the ROW cost is negligible. Due to this, a multiplier of 0.69 is used for the material cost, and a multiplier of 1 is used for the labor and miscellaneous costs assumed in this work.

The pipeline O&M is calculated from a Southern California Gas direct testimony to have their proposed 2019 O&M expenses approved by the CPUC [50]. Dividing their 2016 value of USD 17.7 million for their 3455 transmission pipeline miles for a value of roughly 5100 USD/mi-year.

### 2.3.2. Transmission Compressor

Zhao and Rui [51] consider the construction cost of natural gas compressor stations based on historical costs. They find that the average cost per power capacity of compressor stations is approximately 2800 USD/kW in the Western US region. After accounting for inflation with the United States' Consumer Price Index, this results in approximately 3300 USD/kW for the entire compressor station in 2017. The equations for compressor energy efficiency, compression stages, compressor power rating, and energy consumption are modeled in this work as done in the HDSAM [27].

The United States Environmental Protection Agency [52] utilizes industry data to report that across all natural gas segments, $1.4 \pm 0.5\%$ of the gross natural gas production is lost as emissions. Of this amount, approximately 37% come from the transmission and storage segments, and the measured emissions from pipelines make up less than one thousandth. This implies that pipeline leakage is negligible even if hydrogen would leak at a faster rate [53,54]. Consequently, 1.55% (37% of 1.4% times a factor of 3 for

hydrogen) leakage is modeled to represent the amount of emissions from pipeline and compressor facilities. Leakage in underground storage is modeled to be 0.1% [55].

### 2.3.3. Underground Storage

Lord et al. [56] calculate the levelized cost of storing hydrogen in underground geological features and identify that depleted oil and gas reservoirs are the most economical choice and geographically available in California. For the analysis of this paper, it is assumed that only these depleted oil and gas reservoirs storage types are utilized and costs are calculated as conducted in Lord et al. [56].

Underground storage injection is driven by a compressor while the withdrawal is driven by a high pressure expansion and managed by regulators which are assumed to be negligible regarding cost and energy efficiency (see Figure 2). Amid et al. [57] report using injection pressure between 725 and 1450 psi to store hydrogen in a natural gas reservoir. For the analysis of this paper, it is assumed that the storage compressor outlet average is 1015 psi.

### 2.3.4. Linepack

Linepack is the amount of gas that can be stored in pipelines without surpassing the maximum pressure of 1500 psi and maintaining a minimum pressure of 500 psi. A pipeline wall thickness of 15 mm [58] is assumed, and hydrogen is treated as an ideal gas. A constant temperature of 298 Kelvin and 1000 psi fluctuation allowance is assumed. The constant demand scenario considers all parallel pipelines, whereas the constant transmission utilization factor scenario only has one pipeline. Table 1 summarizes the electric and hydrogen pathway component assumptions, where any models represented by a mathematical equation are tabulated as "Calc" and detailed throughout the text. The fuel cells, gas turbines, and BESS have a high and low value representing current and near-term future capital costs to help bound the results. A summary of the major components is illustrated in Figure 2 for the hydrogen pathways with each labeled symbol modeled with an average efficiency and cost, numerically tabulated in Table 1.

**Table 1.** Pathway major components summary.

| Component | Cost (USD) | Lifespan (Years) | η |
|---|---|---|---|
| Fixed PV Generation Site w/Boost Converter (USD/kW) | 1030 [37] | 30 | 0.95 [43] |
| PV Site Inverter (USD/kW) | 47 [42] | 30 | 0.95 [59] |
| Electrolyzer (USD/kW$_{out}$) | 600–800 [60] | 12 | 0.71 [61] |
| Compressor Substation (USD/kW) | 3300 [51] | 20 | Calc. [27] |
| Transmission Pipeline (USD/mi) | Calc. [26] | 30 | 0.99 [62] |
| Underground Storage Site (USD/kWh) | Calc. [56] | 30 | 1 [56] |
| Fuel Cell (USD/kW) | 2800–3500 [63,64] | 10 | 0.60 [65] |
| Gas Turbine Power Plant (USD/kW) | 1000–1150 [66,67] | 20 | 0.60 [68] |
| 230 kV Double Circuit Power Line (USD/mi) | 4,495,000 [69] | 30 | Calc. |
| 500 kV Double Circuit Power Line (USD/mi) | 9,382,000 [69] | 30 | Calc. |
| 230 kV Substation Base Cost (USD) | 17,710,000 [69] | 35 | N/A |
| 500 kV Substation Base Cost (USD) | 36,194,000 [69] | 35 | N/A |
| 230/66 kV Power Transformer (USD/kVA) | 15.6 [14,45,69] | 40 | 0.97 [42,43] |
| 500/230 kV Power Transformer (USD/kVA) | 16.7 [14,45,69] | 40 | 0.97 [42,43] |
| Li-Ion Energy Storage System (USD/kWh) | 250–350 [69,70] | 20 | 0.69 [46] |

## 2.4. Dynamics and Cost Calculations

We use the California Independent System Operator (CAISO) hourly-resolved aggregated generation and demand data [71]. These profiles for each transmission type and level are scaled independently for each scenario pathway. The constant demand scenarios have an annual peak load of 1200 MW and the solar capacities are varied with storage to meet load. Another set of scenarios is considered where the utilization factor of the transmission medium is held constant at 18% and demand is varied instead. In these scenarios, the demand and generation are scaled simultaneously. Pipeline diameters of 24 inches and 36 inches are considered for the hydrogen transmission and 230 kV, and 500 kV high voltage alternating current (HVAC) levels are considered for the all-electric pathways. Higher transmission levels (i.e., 765 kV HVAC and 42-inch pipelines) are not considered in this analysis due to the sparsity of projects in the region and a lack of data, though they may be more appropriate for extreme transmission distances.

The constant utilization factor scenarios focus on comparing the transmission medium due to differing efficiencies and meet the demand profile scaled by different amounts. This is conducted to compare the two different transmission modes at similar utilization levels to note the impact on LCOT. On the other hand, the constant demand scenarios focus on comparing the entire pathway when a fixed amount of load must be met. All considered scenarios are representative of an actual electric load being met by a set of mostly solar generators complemented by storage. In the constant transmission utilization factor scenarios, there are many nights in which the night-time demand is not met by solar or stored energy for both the all-electric and hydrogen pathways, as it is assumed that other resources in the system can be dispatched (e.g., natural gas combined cycle plants, hydropower). However, in the constant demand scenarios, the entire load is met by the modeled solar or storage throughout the entire year for the hydrogen pathways. For the all-electric pathways, a 95% and 100% of load scenario is presented due to the overall cost sensitivity to the BESS capacity. The storage capacity sufficient for the entire year, corresponding to 100% of the load being met is referred to as the seasonal shifting capacity. The energy storage capacity used to meet only 95% of the load in the all-electric scenario is referred to as the daily shifting capacity.

Substations, transformers, compressors, and electrolyzers are assumed to have a fixed O&M equivalent to 4% of the capital cost. All other components' fixed and variable O&M costs are taken from other sources. The equivalent annual cost (EAC) for each component is the sum of the capital cost divided by annuity factor plus the annual O&M cost. The LCOE is calculated by dividing the total EAC by the delivered electricity meeting load, $E_{throughput}$, as seen in Equation (1) below. A 5% discount rate (DR) per year period is assumed for the annuity factor calculation as shown in Equation (2) below, where n represents the lifespan of each individual component. The LCOT is calculated the same as LCOE but omits the components used for production (i.e., PV solar and electrolyzers). For the hydrogen scenario, the lower heating value of hydrogen is used to quantify the amount of energy being delivered through each component in the pathway in the electrical load scenarios. The LCOT scenarios use hydrogen's higher heating value. Optimal sizing of additional pipelines requires needless complications for the analysis, so all parallel pipelines are the same size.

$$LCOE = EAC/E_{throughput} \tag{1}$$

$$EAC = (CAPEX \times DR)/(1 - (1 + DR)^{-n}) + O\&M \tag{2}$$

## 3. Results

### 3.1. Levelized Cost of Transmission

The highest transmission medium utilization factor in the considered scenarios is 29%, corresponding with the capacity factor of solar in California. In this analysis, the low utilization reflects only serving solar generators. The available transmission capacity to surpass 29% would require different types of additional generators (e.g., wind). Energy

from storage systems meeting demand during non-solar production hours does not contribute to the transmission medium utilization factor as these facilities are downstream of bulk transmission. In general, the hydrogen scenarios show an LCOT that increases at higher utilization due to compressor work, whereas the all-electric pathway LCOT curves increase at longer distances due to ohmic losses. The LCOT metric is the quotient of the transmission components' EAC and the amount of energy available after bulk transmission and storage efficiency losses. Note that the LCOT is the best metric for estimating meeting heat demand, where electric resistive heating is 100% efficient and combustion efficiency is essentially 100%. This LCOT is less complex than that for delivering electricity, which requires accounting for re-electrification and associated losses for the hydrogen transmission pathways. This LCOT is also less complex than that for explicitly meeting hydrogen demand, which requires accounting for electrolysis and associated losses for the all-electric transmission pathways.

LCOT results for all of the scenarios are summarized in Figure 3. At 29% utilization, each pathway and both transmission levels have comparable LCOT in the 100-mile scenario as shown in Figure 3a. As distance grows, higher voltage transmission and larger pipelines are economically better for the accompanying higher throughput. In the 500-mile scenario (Figure 3b), the 36-inch pipeline and 500 kV float at LCOT at approximately 40 USD/MWh, with the electric scenario having lower costs above 25% utilization, otherwise the pipeline scenarios are more effective at lower utilization. For the 900-mile distance analysis (Figure 3c), the pipeline scenarios maintain a slight cost advantage over wire transmission even in the high utilization region as the ohmic losses at 500 kV become more significant. At higher utilization factors, the 230 kV line LCOT increases due to higher power dissipation relative to the 500 kV line. In other words, the higher voltage results in better efficiency by achieving less heat loss and this is most evident in extreme transmission distances.

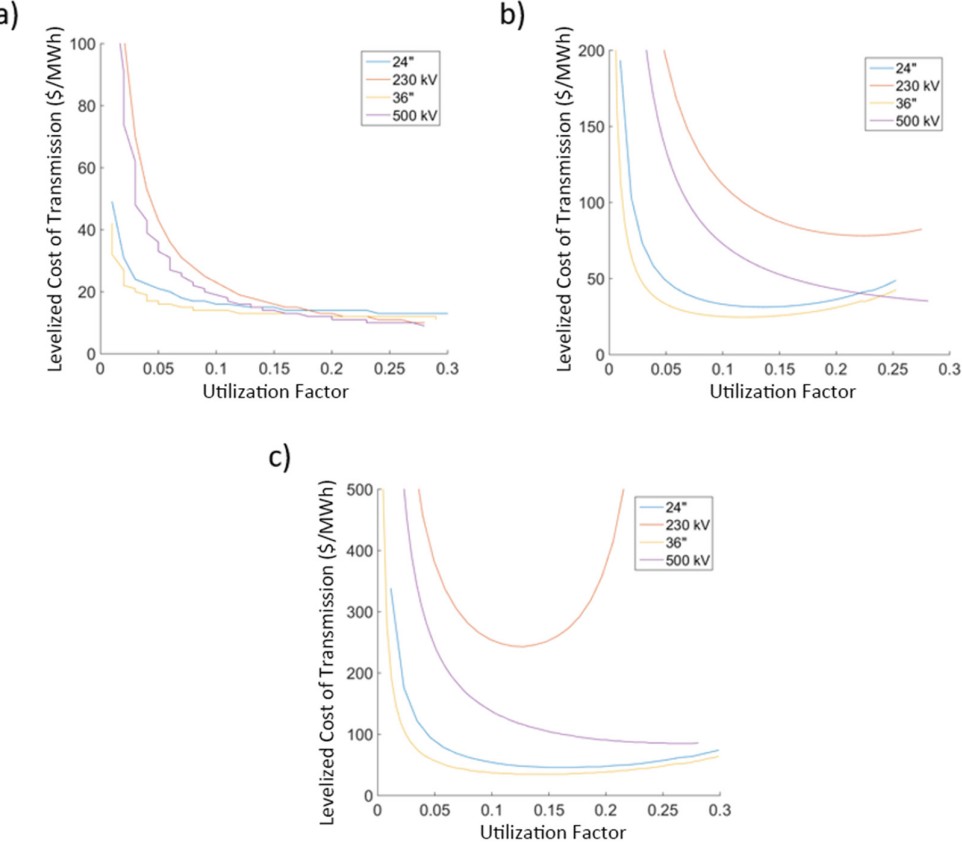

**Figure 3.** Levelized cost of transmitting energy versus utilization factor for (**a**) 100 miles, (**b**) 500 miles, and (**c**) 900 miles.

Adding more upstream solar results in the need to install parallel power lines or pipelines to handle the additional peak power transmission. This binary addition of incremental parallel infrastructure would split the energy throughput between additional lines and lower the utilization factor per line. Because of this, one cannot extrapolate how the curves would behave at higher capacity factors without properly modeling the dynamics of other types of generation and additional interconnections closer to the load.

The transmission power line makes up two-thirds of the all-electric pathway LCOT at 100 miles and increasingly more at longer distances (e.g., 88% at 500 miles in the 500 kV scenario and 90% at 300 miles in the 230 kV scenario). On the other hand, the compressor station components of the gas transmission system consistently remain 73% to 84% of the hydrogen pathway LCOT across the simulated distances.

### 3.2. Levelized Cost of Electricity (LCOE)

Note that in this work LCOE refers to the amount of electricity before the point of sub-transmission. In other words, this metric includes the cost of production, bulk transmission, and preparation and delivery to electric distribution and end-use infrastructure (which is assumed to be identical for all scenarios, but, not included in the cost numbers). The results in this section are presented for two major scenarios: one in which the transmission utilization factor is held constant at 18% and the other in which the same demand profile with a peak 1200 MW demand is met across different transmission medium types and distances. The constant utilization factor scenario attempts to focus on the resulting LCOE with an attempt to provide a comparable analysis on the basis that the transmission component is operating at proportional capacities relative to their investment. The 1200 MW peak demand scenario disregards the transmission utilization factor and focuses on comparing the pathways on the basis that they meet the same end-use electric demand dynamics. Major numerical results for all of these scenarios are tabulated in the Supplementary Materials.

Figure 4 presents a summary of the LCOE results for all of the scenarios considered. The 18% utilization factor scenario results in an LCOE range of 348 USD/MWh to 447 USD/MWh for the all-electric pathway. At 100 miles at 230 kV, the BESS accounts for nearly 80% of the EAC, while the PV accounts for 16% and the transmission line accounts for 3%. At higher transmission distances, the transmission line and PV share grow from increased capacities and circuit-miles but also as the size of the BESS shrinks due to higher transmission losses. This is because in the constant utilization factor scenario, the end load does not impact the storage sizing and only the energy available downstream of the transmission does. To re-iterate, different amounts of demand are being met between the constant utilization and constant demand scenarios. At 100 miles, 230 kV is better at an LCOE of 348 USD/MWh compared to 500 kV's 380 USD/MWh. This is due to the additional transformer cost and energy losses in the 500 kV scenario. The two become more comparable at 300 miles as their LCOE are within 3% of each other, with the 230 kV level still having a slight edge. The 500-kV voltage level becomes more economical beyond this distance.

Note that the cost of the BESS composes most of the LCOE. If we take 20% of the previously discussed 348 USD/MWh to 447 USD/MWh range, this suggests the LCOE without storage in these scenarios can be estimated to be 70 USD/MWh to 90 USD/MWh. This LCOE range is still on the higher end of the 0 USD/MWh to 79 USD/MWh range of the transmitted renewable wind power LCOE found by Mills et al. [72]. However, there are multiple explanations for why their considered transmission projects achieve lower LCOE: (1) the assumed capacity factor for wind is 35% compared to our solar capacity factor of 29%, (2) for transmission projects with known knowledge of other generators, the cost assigned to the wind generator is capacity-weighted, and (3) some of the review transmission projects span more than a thousand miles that (4) are mostly located in rural areas. This suggests that, corresponding to the previous points, (1) the per connected MW-capacity of renewable wind, there is more electricity produced per year than solar to reduce the LCOE that is also (2) cost shared by other generators connected to the line. In addition, (3) our analysis

suggests the modeling of additional transmission lines that specifically access remote regions of bountiful renewables, and these types of lines will typically be shorter than the large backbone transmission lines that connect multiple regions and (4) have higher fractions closer to the load, thus higher suburban development costs. Ultimately, lower LCOE can be achieved by modeling generators outside of solar generation, but the focus of this work is to compare integrated electric and hydrogen energy transmission.

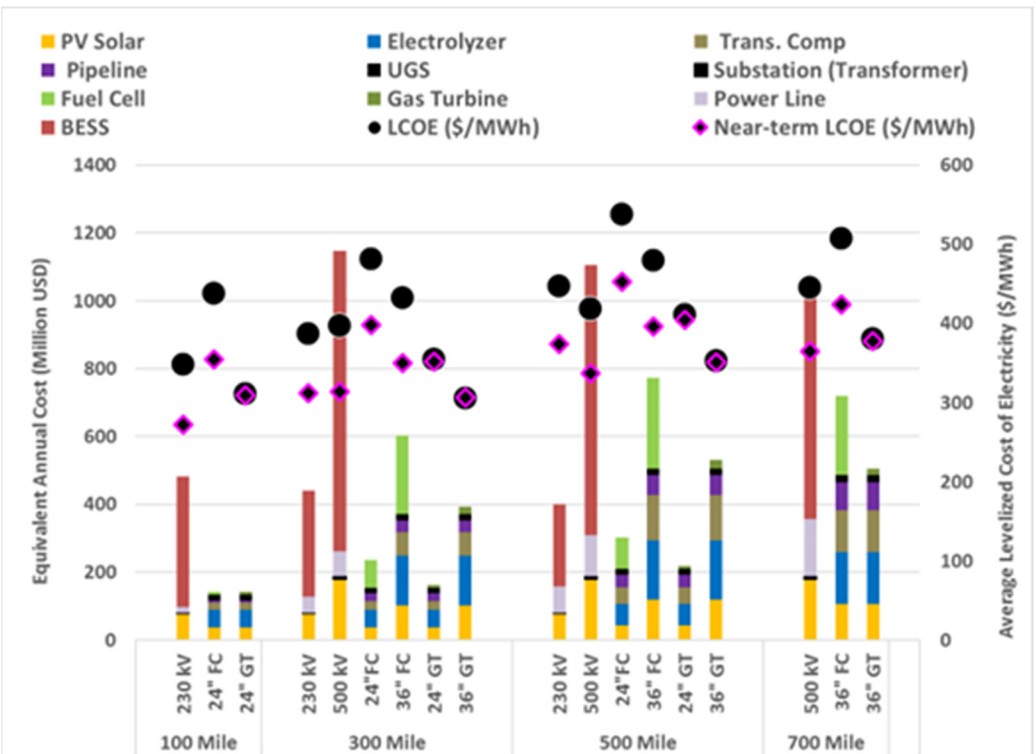

**Figure 4.** Equivalent annual cost breakdown and levelized cost of electricity. Scenario is for constant transmission utilization factor scenario in which the all-electric pathway only meets 95% of load and hydrogen pathways meet 100% of load.

In the hydrogen pathways, the component EACs are more evenly distributed, with only the transmission pipeline percentage of total EAC being in the single digits for transmission distances up until approximately 500 miles. Across the board, the electrolyzer EAC is consistently 46% more than the PV solar EAC as changing the transmission characteristics does not affect these two components' capacity ratio. The hydrogen pathways that include a fuel cell to produce the electricity at the end of transmission span 403 to 538 USD/MWh. The pathways that include a gas turbine for this purpose span 276 to 411 USD/MWh. Using a near-term reduced capital cost for fuel cells results in an average 18% reduction of LCOE across all scenarios. The EAC from each component and the resulting LCOE are summarized in Figure 4. The stacked bars on the graph depict the lower capital cost assumptions that are made to reflect today's costs and the corresponding LCOE is depicted as circles. The LCOE using the lower capital cost assumptions for the electrolyzer, fuel cell, gas turbine, and BESS, made to represent short-term cost reductions from technology maturation, are depicted as diamonds. Note the reduction in the gas turbine scenarios is slight because it is relatively mature technology, so that the symbols overlap.

For both the constant utilization factor and constant demand scenarios, the general trend is that the all-electric pathway remains economically advantageous over the hydrogen transmission pathways that utilize a fuel cell. On the other hand, if the 350 USD/kWh BESS capital and installation cost assumption is made, the hydrogen pathways that utilize a gas turbine are effectively cheaper. Even for the lower 250 USD/kWh BESS capital cost

assumption, the hydrogen gas turbine pathways' resulting LCOE are at most 7% greater than the all-electric scenario.

The following scenarios focus on the constant demand scenarios with a peak load of 1200 MW. BESS can become unreasonably expensive when large energy storage capacities are necessary. As such, a scenario in which only 95% of the load is met for the all-electric profile (hydrogen scenarios are always able to meet 100% of load) is considered in addition to the 100% scenario. The 95% scenario is selected based on the marginalized contribution to meet load despite increased energy storage capacity as shown in Figure 5.

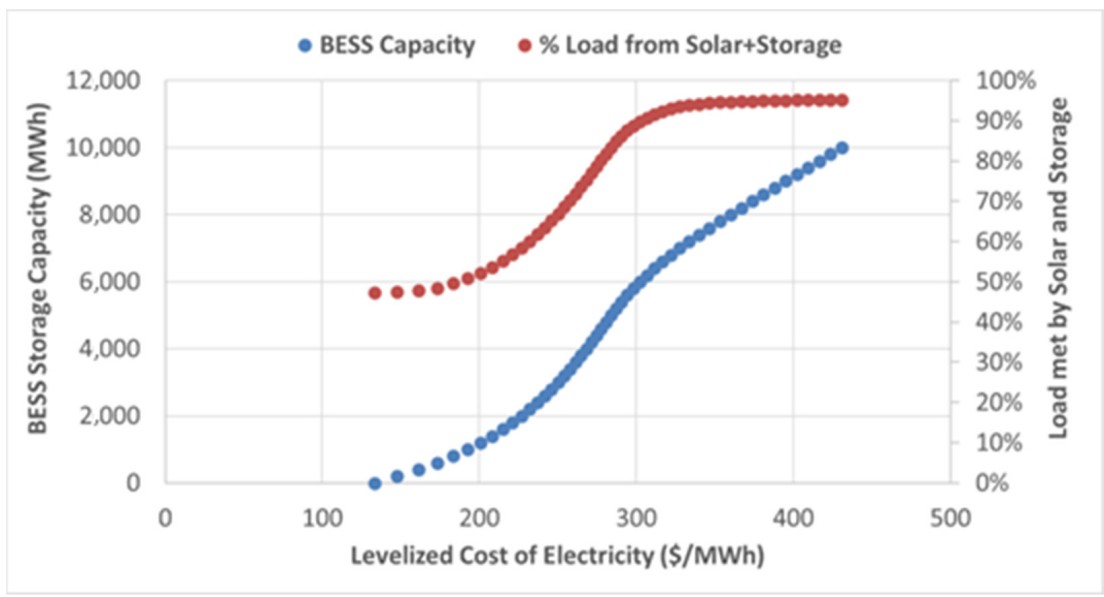

**Figure 5.** Effects on levelized cost of electricity and percentage of load met by increasing battery energy storage capacity in the 100-mile 230 kV scenario.

In the 95% demand scenario, the all-electric pathways' LCOE span from 356 to 455 USD/MWh, while the hydrogen fuel cell pathways span from 389 to 526 USD/MWh, and the hydrogen gas turbine pathways span from 262 to 443 USD/MWh, as shown in Figure 6. Many of the trends previously discussed in the constant transmission utilization factor scenarios are also exhibited in these scenarios as well. The constant demand scenarios allow for a quicker overview of system investment from the consumer perspective as these results are normalized by electric load met. The EAC from each component and the resulting LCOE are presented in Figure 6. Due to dynamics, only 46% and 38% of the transmitted energy is used directly in the constant demand hydrogen and all-electric pathway scenarios, respectively, and the remainder must be sent to storage or curtailed. Storage located downstream of transmission results in differing capacities due to the difference in the discharge efficiencies and self-discharge rates. If storage is modeled upstream of transmission, the storage capacity would be larger in proportion to the transmission losses.

By modeling the fuel cell efficiency as 60%, the discharge efficiency of the battery is still higher at 87%. Le Duigou et al. explore the cost associated with implementing large scale underground hydrogen storage in France and find that in the most demanding scenario (electrolysis driven by purely wind generation), the storage component only accounts for 2.9% of the cost to produce and meet projected transportation demand loads [73]. This is in reasonable agreement with the result of this work as it is found that underground storage accounts for 2% of the pathway cost in the 100-mile scenario—comparable to the 124-mile scenario considered by Le Duigou et al.

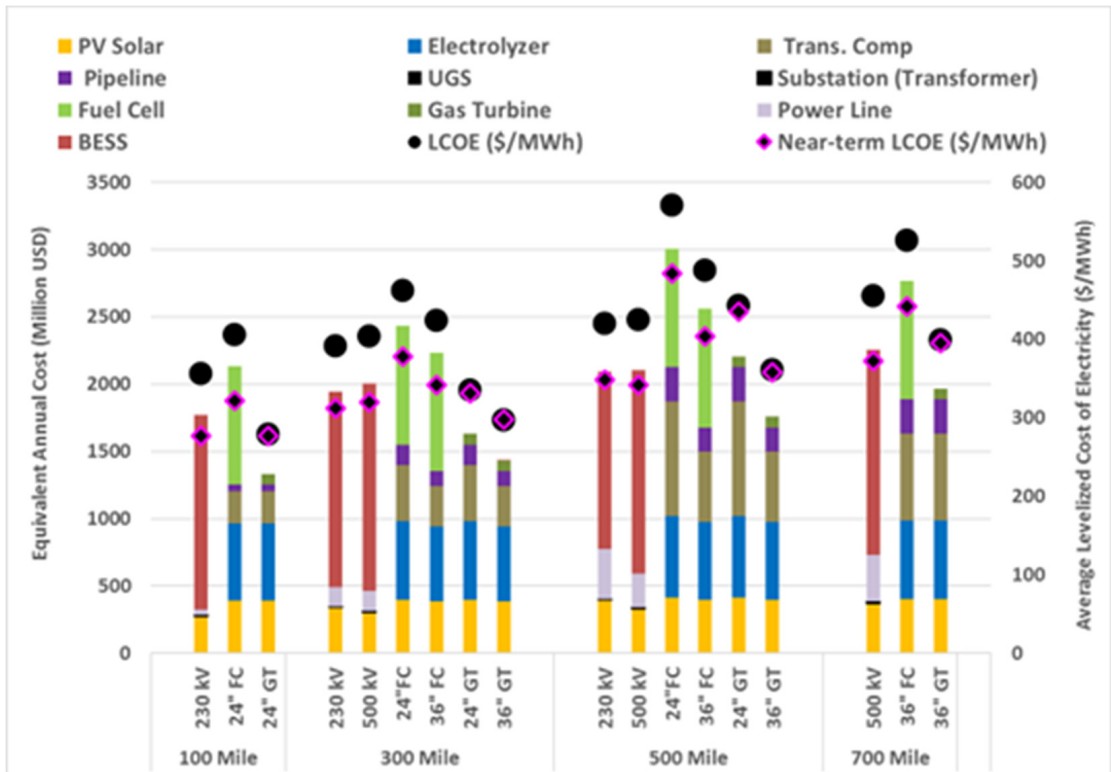

**Figure 6.** Equivalent annual cost breakdown and levelized cost of electricity. Scenario is for the 1200 MW peak demand scenario in which the all-electric pathway only meets 95% of load and hydrogen pathways meet 100% of load.

Past 300 miles, the cost difference between 24-inch and 36-inch pipelines becomes more evident at these throughputs. While the capital cost of the pipelines between the two sizes contributes to a fairly consistent percentage of the total EAC, the transmission compressor grows from 11% for the 100-mile scenario to 28% in the 500 mile-scenario, whereas the 36-inch pathway only increases from 10% to 20% for the same two distances. The amount of linepack from larger pipelines adds another benefit to consider for using larger pipelines at potentially shorter transmission distances.

When comparing the constant utilization and constant demand scenarios, the LCOE for each pathway remains largely similar. Cost differences arise from the demand needing to be scaled to varying amounts in the constant utilization factor scenarios. In addition to the different capacities of solar that would result in an 18% utilization, transmission efficiency is dependent upon transmission type, level, and distance. As such, in the constant utilization factor scenarios, differing capacities of fuel cells must be installed for each scenario. The 36-inch pipeline scenarios have larger required fuel cells as there is more hydrogen available to consume—enabling a larger demand to be met. When normalized by the installed solar capacity, the fuel cell capacity in the 36-inch pipeline scenario is roughly 5% higher than in the 24-inch pipeline scenario. A similar trend is seen in the 500 kV all-electric pathway, where the BESS is sized larger than in the 230 kV pathway to capture greater amounts of available electricity at the end of transmission. For the 100, 300, and 500-mile transmission, the BESS storage capacity, normalized by the installed solar for the 500 kV scenario compared to the 230 kV, increases by 5, 18, and 30%, respectively. These differences ultimately lead to a higher pathway efficiency and illustrate the economy of scale from larger transmission capacities. The transmission utilization factor in the constant demand scenarios span from 14% to 21% in the all-electric scenarios and span from 20% to 28% in the pipeline scenarios.

The LCOE increases with distance due to the compressor and pipeline costs for the hydrogen scenarios and due primarily to power line costs in the electric scenarios. Electrical

component costs increase primarily due to higher power ratings needed, while only pipeline and compressor costs increase in the hydrogen scenario. At higher transmission distances, the hydrogen pathway efficiency is higher due to intermediate compressors working more efficiently. Regarding costs, Table 2 summarizes how each major component cost changes at different distances. As expected, many of the endpoint equipment sizes and costs do not change significantly with transmission distances. For the hydrogen scenario, the compressor cost increases at similar rates as the pipeline itself.

**Table 2.** Comparing change in component EAC when increasing transmission distance. Numbers are provided as a percentage of the shortest transmission distance scenario for (**a**) the electric scenarios and (**b**) the hydrogen scenarios.

| (a) | | | | |
|---|---|---|---|---|
| **Transmission Miles** | **PV** | **Substation & Transformers** | **Power Line** | **Battery** |
| 230 kV Power Line Scenario | | | | |
| 100 | 100% | 100% | 100% | 100% |
| 300 | 109% | 103% | 473% | 100% |
| 500 | 118% | 120% | 788% | 98% |
| 500 kV Power Line Scenario | | | | |
| 300 | 100% | 100% | 100% | 100% |
| 500 | 106% | 102% | 167% | 100% |
| 700 | 115% | 105% | 233% | 100% |

| (b) | | | | | | |
|---|---|---|---|---|---|---|
| **Transmission Miles** | **PV** | **Electrolyzer** | **Compressor** | **Pipeline** | **Fuel Cell** | **Underground Geological Storage** |
| 24-Inch Pipeline Scenario | | | | | | |
| 100 | 100% | 100% | 100% | 100% | 100% | 100% |
| 300 | 100% | 100% | 160% | 299% | 100% | 96% |
| 500 | 100% | 100% | 278% | 499% | 100% | 92% |
| 36-Inch Pipeline Scenario | | | | | | |
| 300 | 100% | 100% | 100% | 100% | 100% | 100% |
| 500 | 98% | 98% | 172% | 167% | 98% | 99% |
| 700 | 98% | 98% | 205% | 233% | 98% | 99% |

Up until this point, the results discussed for the all-electric pathway were for the 95% of load. Despite the all-electric pathways having a higher discharge efficiency, the storage capacity required for the all-electric pathways increases more compared to the hydrogen pathways due to the modestly modeled 4% hourly-resolved self-discharge per month for the BESS. In addition, the increased storage capacity in the BESS scenario requires the purchase of more batteries, which are much more expensive than increasing the size of the storage component (underground storage facilities) only in the hydrogen and pipeline pathways. The increase of BESS capacity required to meet 100% of load scenarios translates into two orders of magnitude increase in LCOE as shown in Figure 7—shifting from the 0.3–0.5 USD/kWh range to a 21–22 USD/kWh range. Note that even if battery costs were to decrease from 350 USD/kWh to 250 USD/kWh, the resulting LCOE would decrease, at most, proportionally. In the hydrogen scenario, the storage capacity costs are associated with underground storage and compressor costs, which on a US dollar per energy storage capacity basis is two orders of magnitude cheaper than battery ESS. The LCOE in the

hydrogen scenarios are already in the 0.3 to 0.6 USD/kWh range, regardless of whether daily or seasonal shifting is necessary. Note that this is due to the low per unit energy storage capacity cost in conjunction with storage already being a small contributor to the total pathway EAC. Note that the significant differences in LCOE between the pathways is illustrated with a logarithmic scale as presented in Figure 7.

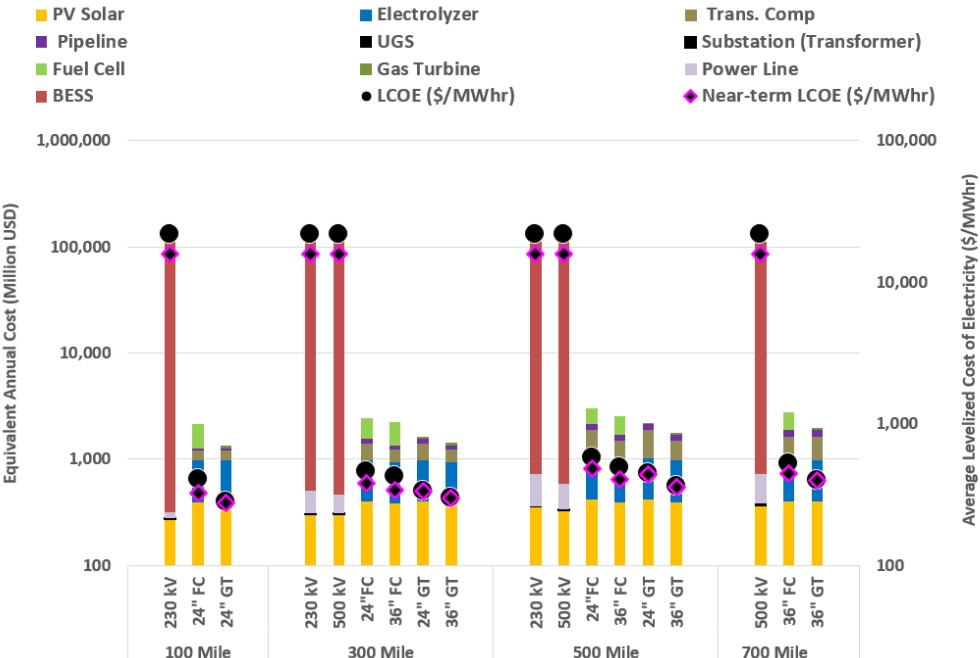

**Figure 7.** Equivalent annual cost breakdown and levelized cost of electricity scenario is for 1200 MW peak demand scenario, in which the all-electric pathway meets 100% of load.

Seven 24-inch and three 36-inch parallel pipelines were necessary in their respective constant demand scenarios. Although it may seem nonintuitive to have three pipelines alongside one another, the model does not physically site the PV installations. As a result, the parallel pipelines are a fair representation of transmitting solar flexibly from multiple generation sites rather than a single location. This idea of incrementally procuring resources suggests that the optimal pathway could change depending upon previously adopted resources and the portion of load being met. The LCOE difference between the all-electric scenario and the hydrogen gas turbine scenario is comparable, depending upon BESS cost reductions and transmission distance in the 95% of load scenario. However, the LCOE for meeting 100% of load indicates the necessity of gas transmission and re-electrification for renewable power to meet the most marginal remainder of load due to techno-economic constraints. This suggests that to meet explicitly electric loads, a combination of the considered pathways could be adopted. The overall makeup of energy demands (i.e., power generation, heating, transportation fuel, and industrial applications) would need to be considered to estimate the balance of adopted pathways.

The amount of energy as hydrogen that could be stored in pipelines is presented in Figure 8. The storage magnitudes are compared to the daily (95%) and yearly (100%) required load-shifting capacity needed in the considered constant demand scenarios. Note that a relatively short pipeline segment could fulfill the daily load-shifting needs, and underground storage would not be necessary for the hydrogen pathway if only daily storage were required. If underground geological storage is to be entirely displaced or is not available in certain locations, then additional pipelines (in addition to those necessary to satisfy the transmission capacity) could be installed.

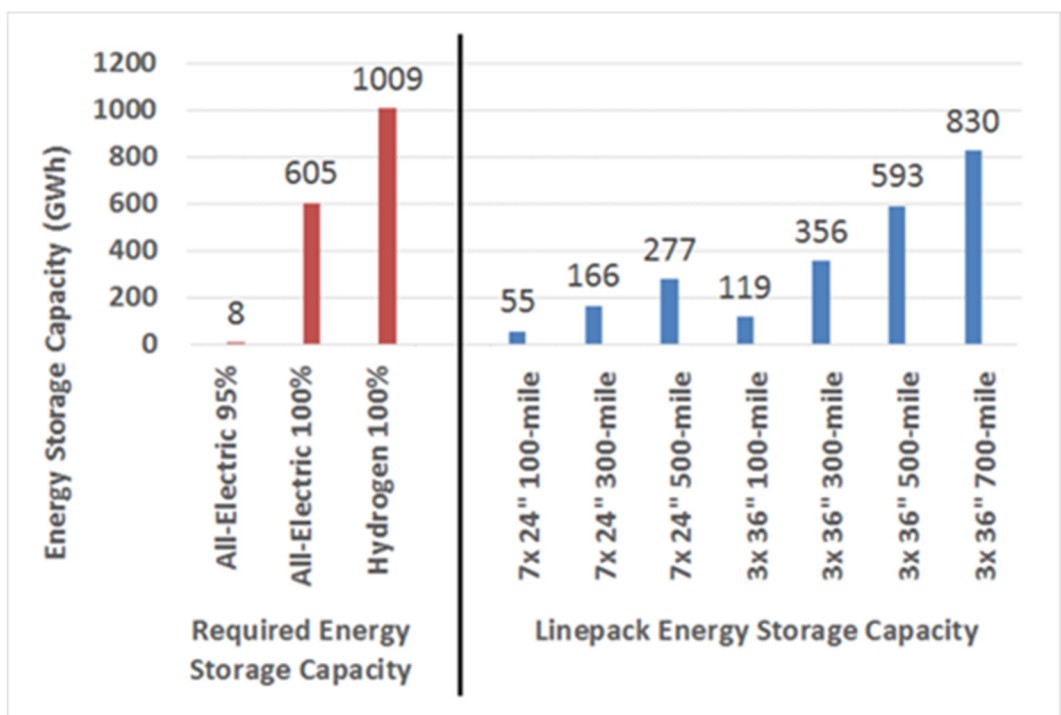

**Figure 8.** Comparison of shifting energy storage capacity for each respective pathway (red) and the amount of pipeline linepack (blue) to meet the 1200 MW peak demand profile.

## 4. Discussion

### 4.1. Transmission Congestion

Many people critique the need for such expansive seasonal storage, but the seasonality of PV solar production is inevitable. This typically results in many hours of curtailment when the load is low or local transmission congestion arises and some long-duration periods in which no solar energy is available. Congestion results in a less profitable market for new developers and may deter deployment. Candidate PV sites and renewable projects are also most commonly located remotely from major loads. As a matter of fact, dense urban environments could never install enough local solar PV to meet energy demands due to insufficient land or rooftop availability in comparison to the energy demands. These remote power producers, who would like to connect to the California grid today, end up often becoming financially responsible for the interconnection costs to the CAISO network. If one were to imagine an equivalent scenario in a hydrogen future, new power producers would be responsible for producing hydrogen on site and for financing the spur pipelines to be integrated to the greater gas grid. Storage systems that are implemented into the grid can act as transmission infrastructure deferral vehicles but also act as energy shifting systems providing further value that is not quantified in this analysis.

Suppose an independent PV farm exports power onto the grid and grid operators are responsible for ensuring that there is available infrastructure to manage this additional power. In the all-electric scenarios, batteries would be responsible for these tasks. Assuming both types of utility networks existed, electrical and gas, batteries or pipelines could be contracted to act as a storage medium. To prevent massive curtailment and ensure the continual deployment of renewables, the amount of energy stored in the pipeline can be considered as the avoided EAC from procuring batteries.

Referring to Figure 8, a pipeline in Central California to a major load, such as Los Angeles, could have sufficient storage capacity for the solar power it delivers. A pipeline to deliver gas to major loads in Los Angeles or towards the major cities in Northern California would allow for many third parties to inject hydrogen produced by excess renewables (i.e., increasingly abundant residential rooftop PV or other utility-scale power

plants) and withdrawn at a later time to meet night-time loads. This would be a practical manifestation of the linepack storage value being realized and increasing total transmission asset utilization.

### 4.2. Reliability and Safety

Overhead power lines have proven to be a hazard in California due to the significant number of wildfires, heat waves, earthquakes, high winds, and other severe weather conditions that the state experiences. The years 2019 and 2020 have been challenging years for Pacific Gas and Electric (PG&E) due to multiple incidents regarding power lines causing fires and public safety power shutoff (PSPS) events, leading them to declare bankruptcy. Underground power lines are an option that would have the advantage of having some protection from severe weather, but the cost can be an order of magnitude higher [11]. In this analysis, it is found that the LCOT is similar for the traditional overhead HVAC to hydrogen pipelines, so that an increase in electric pathway costs for underground reliability and safety should increase the appeal of the hydrogen pathway because it is much less costly than undergrounding the transmission wires. LCOE, which has been dominated by storage costs in the electric pathway, will most likely see significant changes for longer transmission paths. The 500-mile 500-kV transmission line accounts for 12% of the EAC and an order of magnitude increase in the transmission medium could very well double the LCOE. This would suggest that at the same price, hydrogen could meet 100% of the load with renewable energy, whereas the all-electric pathway would deliver far lower magnitudes of renewable energy for the same natural disaster resiliency level. A study [74] from the Gas Technology Institute suggests that the reliability of the gas grid is orders of magnitude higher than the electric grid.

### 4.3. Scalability

In this analysis, scenarios with a peak load of 1200 MW met by a range of 3280 MW (100-mile, 230 kV scenario) to 4900 MW (500-mile, 24-inch scenario) of installed solar capacity are considered. If one were to scale both the generation and demand by a factor of 50, the amount of PV installation is similar to the 154 to 169 GW of PV required in the high solar scenario of Colbertaldo et al. [75], which analyzed a 100% renewable California electric grid. The resulting peak demand is also similar to the historical state peak of 50.3 GW in 2006 [76]. Maintaining the same dynamics of this analysis, one would also need roughly 50 times the amount of energy storage. In the 95% demand all-electric scenario, this would correspond to roughly 0.4 TWh of batteries or 0.7 TWh of hydrogen storage for the hydrogen scenario. Note that more hydrogen storage than battery storage is required due to lower round-trip efficiency. If one were to make the same assumptions of existing pipelines as in this analysis, the collection of pipelines with 19-inch and above diameters in California [77] represents 930 GWh (0.93 TWh) of the linepack storage. What this suggests is that although there may be a proliferation of solar PV plants with batteries, at some point the system will require long-duration storage as most of the power generation assets become solar PV generators. The opportunity to utilize the existing gas infrastructure poses an interesting value proposition—that of being able to shift away from fossil natural gas use while enabling more PV generation with the significant magnitude of linepack storage, which is virtually free.

## 5. Summary and Conclusions

The current analysis objectively compares solar energy delivery via hydrogen through pipelines and via electricity through power lines. All aspects of each delivery system are included to deliver energy to meet electric demands at the endpoint, designated as the point prior to final electric distribution. PV modules and power conversion systems are needed for both the hydrogen and all-electric pathways. However, electrolyzers, compressors, pipelines, and fuel cells or gas turbines are needed for the hydrogen scenarios, whereas transformers, BESS, and power lines are needed in the all-electric scenarios. The comparison

scenarios include: (1) using the same capacity factor of the transmission medium to meet different demand profiles, and (2) delivering the amount of electricity required to meet the same electric demand profile. In this latter scenario, more PV is needed in the lower pathway efficiency hydrogen scenario which requires greater amounts of energy storage capacity despite the higher self-discharge rate from the BESS in the all-electric pathway. This analysis has determined the following:

1. Levelized costs for energy transmission when delivering only solar PV energy over 100 miles via hydrogen through pipelines and electricity through power lines are comparable at roughly 10 USD/MWh to 15 USD/MWh.
2. At lower transmission medium utilization factors, pipelines are significantly cheaper than power lines for delivering energy (e.g., 25 USD/MWh to 30 USD/MWh for pipelines at 10% utilization over 500 miles as opposed to 72 USD/MWh to 118 USD/MWh for power lines).
3. Hydrogen pathways utilizing a gas turbine system for reconversion to electricity are generally the cheapest for meeting electric demand in all scenarios as low as 311, 278, and 278 USD/MWh delivered 100 miles for the 18% utilization scenario, 95% demand, and 100% demand scenarios, respectively.
4. For meeting 95% of demand, the all-electric scenarios are cheaper than the hydrogen pathways which utilize fuel cells (e.g., 356 USD/MWh for all-electric versus 405 USD/MWh for the fuel cell pathway at 100 miles).
5. For meeting 100% of the demand, the hydrogen pathways are two orders of magnitude cheaper than the all-electric pathways due to a marginal increase in cost for hydrogen energy storage (only the gaseous storage component itself must be enlarged) compared to large increases in cost to install larger numbers of complete battery systems (e.g., 405 USD/MWh to 570 USD/MWh for the fuel cell pathway or 278 USD/MWh to 443 USD/MWh for the gas turbine pathway as opposed to the all-electric pathway range of 21,655 to 21,834 USD/MWh).
6. Pipeline linepack energy storage, 55 and 119 GWh with 24-inch and 36-inch pipelines over 100 miles, respectively, is more than sufficient to meet daily shifting demands, roughly 8 GWh in the all-electric pathway and slightly higher in the hydrogen pathways. The pipeline linepack energy storage of 36-inch pipelines over 500 miles nearly provides sufficient storage to meet seasonal storage requirements, roughly 605 GWh in the all-electric pathway, for the PV energy that it transmits.

**Supplementary Materials:** The following supporting information can be downloaded at: https://www.mdpi.com/article/10.3390/en16041880/s1.

**Author Contributions:** Conceptualization, C.T. and J.B.; methodology, C.T.; software, C.T.; validation, C.T.; formal analysis, C.T.; investigation, C.T.; resources, J.B.; data curation, C.T.; writing—original draft preparation, C.T., J.B.; writing—review and editing, J.B; visualization, C.T.; supervision, J.B.; project administration, J.B.; funding acquisition, J.B. All authors have read and agreed to the published version of the manuscript.

**Funding:** This research was funded by University of California Office of the President as a part of the Carbon Neutrality Initiative Renewable Energy Storage Project contract.

**Data Availability Statement:** Data is contained within the article or Supplementary Material.

**Conflicts of Interest:** The authors declare no conflict of interest. The funders had no role in the design of the study; in the collection, analyses, or interpretation of data; in the writing of the manuscript, or in the decision to publish the results.

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
