# Peer review of "Comparative Levelized Cost Analysis of Transmitting Renewable Solar Energy"

_energies, doi:10.3390/en16041880_

Round 1

Reviewer 1 Report

The article presents an interesting aspect of cost estimation in the power industry. The structure of the article is consistent with the editorial requirements of the journal. However, in the case of conclusions, the statements should be supplemented with values that have been calculated on the basis of the presented analyses.

Reviewer 2 Report

General comment

The authors present a comparative techno-economic analysis of solar energy carriers—hydrogen and electricity, with an emphasis on the levelized cost of transmission and levelized cost of electricity. The paper is of importance to renewable hydrogen economy today. The analysis and results are clearly presented, and the paper is well-written and deserves to be published. My comments are as follows:

1) Figure 3c, authors should explain the increasing trend of LCOT with respect to the high utilization factor when it comes to 230 kV.

2) Consider all-electric mode, at the same time, only excess solar PV energy is used to make hydrogen via electrolysis and save in storage for use when demand is high. Will the combination of all-electric and power-hydrogen-power be more cost-competitive? 

Reviewer 3 Report

This paper compared the LCOT and LCOE among some systems considering the pipelines. I feel this paper could be improved in the following aspects:

(1) Section 2

(i)Before 2.1, some schemetic diagram should be added. Also this part should be improved for reader's better understanding.

(ii)Figure 1 and 2 should be cited and explained in section 2.

(iii)How could be the authors model or simulate the system?

(2) I think the authors may want to compare different systems. Could the author clearly explain system?

(3)The logic of the paper should be further improved.

Round 2

Reviewer 3 Report

(1) In the comment 1 in the last review, I suggested the author to improve the system description. In this version, this part has not been improved.

Actually, there should not be so many words between the headline 2 and the headline  2.1, which makes this part out of logic.

It is suggested to describe the system clearly first then describe each component in detail.  

(2) In the comment 2 in the last review, I suggested the author to explain Figs 1 and 2. However, it is not improved in the new version.

I believe Figs 1 and 2 represents the conversion and transfer pathways of the energy. It is more logical to describe it before the description of major components in detail.

(3)  In the comment 3 in the last review, I suggested the author to add the model or simulation method. 

For example, what is mathematical equation of LCOE or LCOT? Any mathematical equation to represent the energy conversion in each component?

(4) In the comment 4 in the last review, I suggested the author to improve the logic of the whole manuscript. However, it is not improved in the new version. 
